# Components of clean delivery kits and newborn mortality in the Zambia Chlorhexidine Application Trial (ZamCAT): An observational study

Jason H. Park[1]*, Davidson H. Hamer[2,3], Reuben Mbewe[4], Nancy A. Scott[2], Julie M. Herlihy[5], Kojo Yeboah-Antwi[6], Katherine E. A. Semrau[7,8,9]

1 Department of Internal Medicine, University of Michigan Hospitals, Ann Arbor, Michigan, United States of America, 2 Department of Global Health, Boston University School of Public Health, Boston, Massachusetts, United States of America, 3 Section of Infectious Diseases, Department of Medicine, Boston Medical Center, Boston, Massachusetts, United States of America, 4 Levy Mwanawasa University Teaching Hospital, Lusaka, Zambia, 5 Department of Pediatrics, Boston University School of Medicine, Boston, Massachusetts, United States of America, 6 Father Thomas Alan Rooney Memorial Hospital, Asankrangwa, Ghana, 7 Ariadne Labs, Brigham and Women's Hospital and Harvard T.H. Chan School of Public Health, Boston, Massachusetts, United States of America, 8 Division of Global Health Equity, Brigham and Women's Hospital, Boston, Massachusetts, United States of America, 9 Department of Medicine, Harvard Medical School, Boston, Massachusetts, United States of America

* parkjaso@med.umich.edu

**Data Availability Statement:** The data is available at Boston University's data repository: https://open.bu.edu/handle/2144/42348.

## Abstract

### Background

Neonatal infection, a leading cause of neonatal death in low- and middle-income countries, is often caused by pathogens acquired during childbirth. Clean delivery kits (CDKs) have shown efficacy in reducing infection-related perinatal and neonatal mortality. However, there remain gaps in our current knowledge, including the effect of individual components, the timeline of protection, and the benefit of CDKs in home and facility deliveries.

### Methods and findings

A post hoc secondary analysis was performed using nonrandomized data from the Zambia Chlorhexidine Application Trial (ZamCAT), a community-based, cluster-randomized controlled trial of chlorhexidine umbilical cord care in Southern Province of Zambia from February 2011 to January 2013. CDKs, containing soap, gloves, cord clamps, plastic sheet, razor blade, matches, and candle, were provided to all pregnant women. Field monitors made a home-based visit to each participant 4 days postpartum, during which CDK use and newborn outcomes were ascertained. Logistic regression was used to study the association between different CDK components and neonatal mortality rate (NMR). Of 38,579 deliveries recorded during the study, 36,996 newborns were analyzed after excluding stillbirths and those with missing information. Gloves, cord clamps, and plastic sheets were the most frequently used CDK item combination in both home and facility deliveries. Each of the 7 CDK components was associated with lower NMR in users versus nonusers. Adjusted logistic

**Funding:** Bill & Melinda Gates Foundation (Global Health Grant Number OPPGH5298, https://www.gatesfoundation.org/) funded DH. Fogarty International Center (Award Number D43 TW010543, https://www.fic.nih.gov/) funded JP. The funders had no role in study design, data collection and analysis, decision to publish, or preparation of the manuscript.

**Competing interests:** The authors have declared that no competing interests exist.

**Abbreviations:** CDK, clean delivery kit; NMR, neonatal mortality rate; OR, odds ratio; WHO, World Health Organization; ZamCAT, Zambia Chlorhexidine Application Trial.

regression showed that use of gloves (odds ratio [OR] 0.33, 95% CI 0.24–0.46), cord clamp (OR 0.51, 95% CI 0.38–0.68), plastic sheet (OR 0.46, 95% CI 0.34–0.63), and razor blade (OR 0.69, 95% CI 0.53–0.89) were associated with lower risk of newborn mortality. Use of gloves and cord clamp were associated with reduced risk of immediate newborn death (<24 hours). Reduction in risk of early newborn death (1–6 days) was associated with use of gloves, cord clamps, plastic sheets, and razor blades. In examining perinatal mortality (stillbirth plus neonatal death in the first 7 days of life), similar patterns were observed. There was no significant reduction in risk of late newborn mortality (7–28 days) with CDK use. Study limitations included potential recall bias of CDK use and inability to establish causality, as this was a secondary observational study.

### Conclusions

CDK use was associated with reductions in early newborn mortality at both home and facility deliveries, especially when certain kit components were used. While causality could not be established in this nonrandomized secondary analysis, given these beneficial associations, scaling up the use of CDKs in rural areas of sub-Saharan Africa may improve neonatal outcomes.

### Trial registration

Name of trial: Zambia Chlorhexidine Application Trial (ZamCAT) Name of registry: Clinical-trials.gov Trial number: NCT01241318.

## Author summary

### Why was this study done?

- Infection during childbirth is a major cause of newborn mortality and morbidity in rural and resource-limited settings.

- Clean delivery kits can prevent pathogen transmission during infection by providing sterile equipment and encouraging hygienic behaviors.

- A more nuanced understanding of the benefits of clean delivery kits and their components would aid global implementation to help reduce newborn mortality.

### What did the researchers do and find?

- We analyzed the data from the Zambia Chlorhexidine Application Trial (ZamCAT), a cluster-randomized controlled trial conducted in Zambia in 2011–2013.

- During ZamCAT, we provided clean delivery kits to all mothers in the community and tracked the health outcomes of all women and newborns through the neonatal period.

- Analysis of 38,579 deliveries showed us that use of gloves, cord clamps, plastic sheets, and razor blades during intrapartum care were associated with lower newborn mortality, in both home and facility deliveries.

- Components of clean delivery kits were associated with lower risk of perinatal and newborn mortality in the first 7 days of life, but not with mortality between 7 and 28 days of life.

### What do these findings mean?

- Components of clean delivery kits that showed highest usage and association with newborn mortality reduction should be included in future clean delivery kit interventions.

- As this is a post hoc secondary data analysis and the trial was not meant to specifically study the impact of clean delivery kits, causality cannot be inferred.

- Nevertheless, the large sample size, variation in component use, and study design focused on community settings provide support for clean delivery kits as an intervention that may improve newborn health outcomes in both home and facility deliveries.

## Introduction

Despite progress in decreasing child mortality between 1990 and 2015, reductions in neonatal mortality have been slower, remaining unacceptably high at 2.9 million deaths globally each year [1,2]. Infection is one of the leading causes of neonatal death—responsible for up to 35% of neonatal deaths—and 30%–40% of neonatal infections are from pathogens acquired from the mother or environment during labor and delivery leading to neonatal sepsis [3–5]. A disproportionate amount of this burden occurs in low- and middle-income countries.

In order to prevent infection at birth and reduce maternal and neonatal mortality, the World Health Organization (WHO) promotes clean delivery practices through the observance of "six cleans" at the time of delivery: clean hands, perineum, surface, cord cutting and tying, and nothing unclean introduced into the vagina [6]. Clean delivery kits (CDKs) include tools such as soap, gloves, cord ties, and other sterile equipment to enable the "six cleans," offering a potential low-cost intervention to facilitate clean delivery practices [7]. Studies assessing the impact of CDK use have shown a consistent increase in clean delivery practices [8–10]. Studies in South Asia where CDKs were part of a bundle of interventions demonstrated their efficacy in significantly reducing perinatal and neonatal mortality [11,12]. Given the additional evidence for protection against cord infection and neonatal tetanus, CDKs appear to be an effective intervention to help reduce neonatal morbidity and mortality [9,13].

Despite evidence of the CDKs' benefit for newborn health, there are limitations to our current knowledge of their impact. First, there have been few studies of differences in impact of CDK use on immediate, early, and late neonatal mortality. CDK use may have protective effects beyond immediate infection control as CDKs have been associated with reductions in complications from neonatal prematurity [11,12]. Thus, a more comprehensive time to event analysis could elucidate the role of CDKs in preventing varied conditions during the early

newborn period. Second, previous studies noted differences in CDK components used in home as opposed to facility deliveries, but there is no published information about the value each component has in improving neonatal outcomes [10]. Finally, there have not been large-scale studies assessing the impact of CDKs in sub-Saharan Africa. The most robust literature to date has come from South Asia and the Middle East [11–13]. It is reasonable to expect that geographic, demographic, and cultural differences, such as home delivery rates and practices, antenatal care service usage, and prevalence of maternal HIV, are important covariates that may influence perinatal outcomes in sub-Saharan Africa.

Understanding the differential impact of the CDK components and population characteristics that can affect the efficacy of CDKs can inform future program design to implement CDKs. Utilizing data from the Zambia Chlorhexidine Application Trial (ZamCAT), a cluster-randomized controlled trial examining the impact of topical application of 4% chlorohexidine on the cord stump in reducing rates of neonatal mortality in Southern Province of Zambia [14], we investigated the relationship between the individual components of CDKs and neonatal health outcomes for home and facility deliveries.

## Methods

### Study design

ZamCAT was a community-based, cluster-randomized controlled trial in Southern Province, Zambia, that investigated the impact of daily cord care with 4% chlorhexidine on reducing severe omphalitis and neonatal mortality compared to dry cord care. The ZamCAT design and methods have been reported elsewhere; the study was conducted from February 2011 to January 2013 [15]. In the main intent-to-treat and per-protocol analyses, 4% chlorhexidine did not reduce neonatal mortality compared to dry cord care [14]. The present analysis was not in the prespecified analysis plan but was not data-driven. For this secondary, post hoc analysis, the data were pooled across randomization from the initial arms of the study design.

### Study population

The trial was implemented in 6 of the 11 districts in Southern Province—Mazabuka, Siavonga, Monze, Choma, Kalomo, and Livingstone. The study locations included a wide range of health facilities, both urban and rural, in 90 facility catchment areas across the 6 districts. Eligible healthcare centers were Zambian government or mission primary healthcare centers providing routine antenatal services and conducting a minimum of 160 births per year.

At the time of the study, the most recent Zambia Demographic and Health Survey report in 2007 estimated a neonatal mortality rate (NMR) at 34 per 1,000 live births [16]. While more than 90% of pregnant women had an initial registration at an antenatal clinic, only 60% had 4 or more antenatal visits. In Southern Province, 62% of the deliveries occurred at home, 99.3% of which were attended by nonskilled providers, including traditional birth attendants, untrained family, or community members [16]. In comparison, 96.6% of deliveries at health facilities were attended by skilled providers including doctors, clinical officers, midwives, and nurses.

### Eligibility and enrollment

The study field monitors screened women attending antenatal care visits at eligible health centers and during community outreach activities for study enrollment. Eligible women were in their second or third trimester, aged 15 years or older, planned to stay in the catchment area until 28 days postpartum, and provided informed consent. At enrollment, data such as

expected date of delivery and demographic information were collected. Participants were encouraged to deliver at their nearest health facility and were given standard newborn care messages per national guidelines including information about delivery location, breastfeeding, cord care, and danger signs of ill health in their newborn baby. Further information on enrollment procedures is reported elsewhere [15].

## Study procedures

Field monitors made 5 home visits—1 antenatal and 4 postnatal (day 1, 4, 10, and 28 postpartum). The antenatal visit was completed within 2 weeks of enrollment, during which the field monitor confirmed the home location for follow-up, reviewed study procedures with the mother, and screened for pregnancy danger signs. During this initial antenatal visit, field monitors provided a standard CDK to all study participants, regardless of randomization assignment. The CDK, developed in coordination with the Zambia Ministry of Health, contained a bar of soap, sterile razor blade, sterile gloves, 2 cord clamps, candle, matches, and a plastic sheet. The cost of each CDK was about US$5.50. At each of the 4 postnatal visits, field monitors assessed and documented the mother's and newborn baby's health status. At the day 4 postpartum visit, the field monitors asked the mother about use of each component of the CDK. Any mother or newborn baby with danger signs or symptoms, as defined by WHO's signs of newborn possible severe bacterial infection [17], was referred to the healthcare center.

## Variables and outcomes

Each of the 7 CDK components was categorized as a binary variable depending on whether it was used or not, based on maternal self-report at the day 4 postpartum visit. With 7 CDK components, there were 128 potential combinations of use. Every combination of CDK component use was tallied to extract the most common combinations, stratified by delivery location. Delivery location was categorized as "facility" if it was a hospital or health facility, or "home" if otherwise. Maternal age and education were captured. Newborn gestational age was calculated based on weeks since last menstrual period at enrollment and weeks from enrollment to birth. Gestational age was dropped if less than 0 or greater than 42 weeks. Gestational age at birth and birth weight were categorized according to WHO's classification: term (>37 weeks), preterm (32–37 weeks), very preterm (28–31 weeks), and extremely preterm (<28 weeks), and normal birth weight (>2.5 kg), low birth weight (1.5–2.5 kg), and very low birth weight (<1.5 kg) [18,19].

In this analysis, the primary outcome is all-cause neonatal mortality within 28 days postpartum. Deaths were documented by interview with the mother; stillbirth was defined as an infant who did not breathe, cry, or move at the time of delivery. In case of death, verbal autopsies were completed including information on the time of birth, death, and surrounding circumstances. Trained clinicians reviewed the verbal autopsy reports and determined the cause of death. The timing of newborn death was categorized as immediate if in the first 24 hours of life, early if at 1 to 6 days of life, and late if at 7 to 28 days of age [20]. NMR for each CDK component was calculated by using the number of newborn deaths stratified by usage of each CDK component. This was a descriptive exercise, and the statistical significance of differences in NMR was not calculated as the sample size of nonusers of specific CDK components was too low. Similarly, we examined perinatal mortality (stillbirth plus neonatal death in the first 7 days of life) and conducted the analysis in the same manner as described above for neonatal mortality. Signs of infection such as baby's temperature, convulsions/fits, or lack of movement without stimulus were assessed by field monitors at each postnatal visit. For rates of presence

of infection danger signs, newborns were censored if their age of death occurred prior to the field monitor visit for assessment.

## Data analysis/statistical methods

This is a post hoc secondary analysis of data collected in ZamCAT, a cluster-randomized controlled trial. We decided to do this additional analysis for 2 reasons. First, we wanted to understand the uptake and use of CDKs in the population studied during the trial. Second, we wanted to assess the associations between kit use and the health outcomes of newborns.

The main study targeted enrollment of 42,570 women; we used the full cohort for this secondary analysis of the impact of CDK use on neonatal mortality, as the main study did not see a difference in newborn outcomes between the study and control arms [14,15]. Complete case analysis was used, and cases with missing delivery location, CDK use, and newborn outcome were excluded from analysis. We initially excluded stillbirths from the analysis as CDKs would not have had any effect on deaths that occurred antepartum and CDKs may have been less likely to be used if fetal demise was noted or suspected prior to labor. To understand the potential impact on intrapartum stillbirths and address potential misclassification with early neonatal death, we assessed perinatal mortality as an outcome. Bivariate logistic regression was used to compare proportions of neonatal deaths stratified by each CDK component, delivery location (home versus facility), and immediate/early (0–6 days of age) versus late (7–28 days of age) mortality. Each of the 128 permutations of the 7 CDK components was analyzed by frequency of use. The 5 most frequently used combinations at home and facility deliveries were analyzed for their NMR. Baseline characteristics chosen as key covariates were based on the main study, with a focus on known risk factors for newborn sepsis and mortality, including newborn sex, low birth weight, prematurity, maternal age, and maternal education [14]. Given the parent study, chlorohexidine application was considered as a covariate. However, we excluded chlorhexidine use due to survivor bias as ongoing chlorhexidine application was dependent on newborn survival.

Data analysis was performed using Stata 15. The main study was registered at Clinical-Trials.gov (NCT01241318).

## Ethical approval

The Boston University Medical Campus Institutional Review Board (protocol #H-29647) and University of Zambia Research Ethics Committee (protocol #016-08-10) provided ethical approval, and the Zambia Ministry of Health approved the study. All women provided written informed consent, which was obtained in either English or Tonga.

## Results

### Study population

There were 38,579 deliveries recorded during the study. Deliveries with missing information on CDK use, newborn outcome, or delivery location were excluded from analysis ($n$ = 960 deliveries). For the primary analysis, 623 stillbirths were removed, resulting in 36,996 newborns from 36,606 women for analysis (Fig 1). Women were on average 25.6 years old; 83% were married, and 63.6% delivered at a hospital or a health facility (Table 1). Overall NMR was 14 deaths per 1,000 live births, approximately half of which occurred within the first 24 hours after birth. NMR was similar across delivery location (home or facility). While not all descriptive information was available for the excluded participants, overall demographics were similar to those of the included participants.

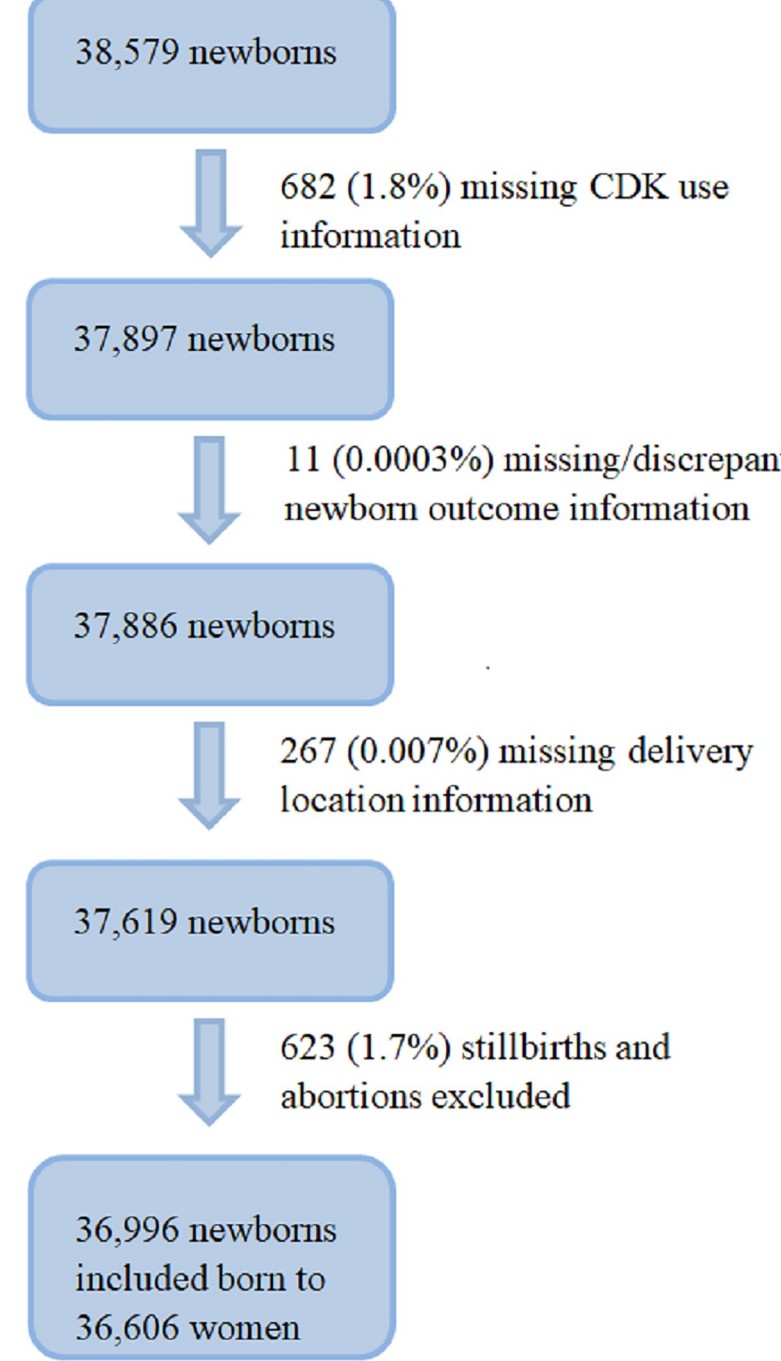

**Fig 1. Participant flow chart.** CDK, clean delivery kit.

## CDK use

CDK use was nearly universal; at least 1 component was used in 99.3% of deliveries, and at least 4 of the 7 components were used in 89.9% of deliveries. All 7 components were used in 28.1% of deliveries. Gloves, cord clamp, plastic sheet, and razor blade were the most frequently used components (Table 2). Women who delivered at home were more likely to have used the

**Table 1. Participant characteristics.**

| Mother characteristics | Number of women (*n* = 36,606) | Excluded women (*n* = 1,583) |
|---|---|---|
| Age (years) | | |
| Less than 20 | 8,519 (23.3%) | 357 (22.6%) |
| 20–29 | 17,548 (47.9%) | 670 (42.3%) |
| 30–39 | 9,079 (24.8%) | 422 (26.7%) |
| 40 or more | 1,460 (4.0%) | 134 (8.5%) |
| Marital status | | |
| Married | 30,190 (82.5%) | 1240 (78.3%) |
| Formerly married | 563 (1.5%) | 23 (1.5%) |
| Other | 5,665 (15.5%) | 255 (16.1%) |
| No response | 188 (0.5%) | 65 (4.1%) |
| Education | | |
| Did not finish primary school | 22,566 (61.7%) | 968 (61.2%) |
| Finished primary but not secondary school | 13,533 (37.0%) | 538 (34.0%) |
| Finished secondary school or more | 303 (0.8%) | 11 (0.7%) |
| No response | 204 (0.6%) | 66 (4.2%) |
| Delivery location | | |
| Hospital | 4,954 (13.5%) | 240 (30.2%) |
| Health center | 18,343 (50.1%) | 280 (35.2%) |
| Home | 13,005 (35.5%) | 259 (32.5%) |
| Other | 304 (0.8%) | 17 (2.1%) |
| **Newborn characteristics** | **Number of newborns (*n* = 36,996)** | **Excluded newborns (*n* = 1,583)** |
| Birth sex | | |
| Male | 18,170 (49.1%) | 675 (55.6%) |
| Female | 18,532 (50.1%) | 540 (44.4%) |
| No response | 294 (0.8%) | 0 (0.0%) |
| Estimated gestational age | | |
| Term (>37 weeks) | 27,828 (75.2%) | 1142 (72.1%) |
| Preterm (32–37 weeks) | 7,718 (20.9%) | 299 (18.9%) |
| Very preterm (28–31 weeks) | 1,002 (2.7%) | 82 (5.2%) |
| Extremely preterm (<28 weeks) | 448 (1.2%) | 60 (3.8%) |
| Birth weight | | |
| Normal (>2.5 kg) | 35,338 (95.5%) | 1554 (98.2%) |
| Low (1.5–2.5 kg) | 1,587 (4.3%) | 27 (1.7%) |
| Very low (<1.5 kg) | 71 (0.2%) | 2 (0.1%) |

soap, razor blade, candle, and matches, while women who delivered in facilities were more likely to have used the cord clamp than those who delivered at home. Gloves, cord clamp, and plastic sheet were used together in the 5 most common combinations of CDK use across all locations, accounting for 85.9% of all combinations (Table 3). The razor blade was used in the top 5 most common CDK combinations at home deliveries, and was used in the top 3 most common CDK combinations among facility deliveries.

The difference in NMR (deaths per 1,000 live births) between use and nonuse of each CDK component showed a reduction in risk for each of the CDK components individually. The greatest NMR reductions were seen for the use of the gloves (nonuse: 84.4 deaths per 1,000 live births; use: 11.9 deaths per 1,000 births), cord clamp (nonuse: 51.3 deaths per 1,000 live births;

**Table 2. CDK use by component.**

| CDK component | Component use, n (%) | | | NMR (deaths/1,000 live births) at all delivery locations | |
|---|---|---|---|---|---|
| | All delivery locations (n = 36,996) | Home (n = 13,412) | Facility (n = 23,584) | Component use | Component nonuse |
| Soap | 27,422 (74.1%) | 11,491 (85.7%) | 15,931 (67.6%) | 11.4 | 21.7 |
| Gloves | 35,858 (96.9%) | 13,029 (97.1%) | 22,829 (96.8%) | 11.9 | 84.4 |
| Cord clamp | 34,795 (94.1%) | 11,851 (88.4%) | 22,944 (97.3%) | 11.7 | 51.3 |
| Plastic sheet | 35,478 (95.9%) | 13,176 (98.2%) | 22,302 (94.6%) | 11.8 | 67.2 |
| Razor blade | 32,453 (87.7%) | 13,087 (97.6%) | 19,366 (82.1%) | 11.3 | 33.7 |
| Candle | 12,802 (34.6%) | 7,321 (54.6%) | 5,481 (23.2%) | 10.3 | 16.1 |
| Matches | 12,588 (34.0%) | 7,250 (54.1%) | 5,338 (22.6%) | 10.2 | 16.1 |

CDK, clean delivery kit; NMR, neonatal mortality rate.

use: 11.7 deaths per 1,000 live births), and plastic sheet (nonuse: 67.2 deaths per 1,000 live births; use: 11.8 deaths per 1,000 live births) (Table 2). Similar results were found when examining the association of CDK use with perinatal mortality. There was a perinatal mortality reduction seen for use relative to nonuse of gloves, cord clamps, plastic sheets, and razor blades (S1 Table). Unadjusted, descriptive data showed that mothers who were younger and highly educated had lower NMR. Newborn characteristics associated with higher NMR included male sex, lower gestational age, and lower birth weight.

## Unadjusted and adjusted logistic regression models

Unadjusted logistic regression showed strong independent relationships between use of gloves, cord clamp, plastic sheet, and razor blade and lower rates of newborn death across all delivery

**Table 3. CDK use by most common combinations.**

| Most frequent CDK combinations | Soap | Gloves | Cord clamp | Plastic sheet | Razor blade | Candle | Matches | Percent used (%) | NMR (deaths/1,000 live births) |
|---|---|---|---|---|---|---|---|---|---|
| All delivery locations | | | | | | | | | |
| 1 | Yes | Yes | Yes | Yes | Yes | No | No | 36.7 | 11 |
| 2 | Yes | Yes | Yes | Yes | Yes | Yes | Yes | 28.1 | 10 |
| 3 | No | Yes | Yes | Yes | Yes | No | No | 12.8 | 13 |
| 4 | No | Yes | Yes | Yes | No | No | No | 4.7 | 15 |
| 5 | Yes | Yes | Yes | Yes | No | No | No | 3.7 | 17 |
| Home deliveries | | | | | | | | | |
| 1 | Yes | Yes | Yes | Yes | Yes | Yes | Yes | 43.3 | 8 |
| 2 | Yes | Yes | Yes | Yes | Yes | No | No | 31.6 | 10 |
| 3 | No | Yes | Yes | Yes | Yes | No | No | 6.0 | 16 |
| 4 | Yes | Yes | No | Yes | Yes | Yes | Yes | 4.3 | 7 |
| 5 | No | Yes | Yes | Yes | Yes | Yes | Yes | 3.9 | 10 |
| Facility deliveries | | | | | | | | | |
| 1 | Yes | Yes | Yes | Yes | Yes | No | No | 39.5 | 12 |
| 2 | Yes | Yes | Yes | Yes | Yes | Yes | Yes | 19.5 | 13 |
| 3 | No | Yes | Yes | Yes | Yes | No | No | 16.7 | 12 |
| 4 | No | Yes | Yes | Yes | No | No | No | 7.2 | 14 |
| 5 | Yes | Yes | Yes | Yes | No | No | No | 5.4 | 16 |

CDK, clean delivery kit; NMR, neonatal mortality rate.

locations. Home deliveries especially had lower rates of newborn death when there was use of gloves, plastic sheets, and razor blades. In facility deliveries, use of gloves, cord clamp, and plastic sheet were each associated with lower likelihood of newborn mortality.

Logistic regression adjusted for other risk factors for newborn mortality showed similar associations of CDK component use with newborn mortality, with gloves, cord clamp, plastic sheet, and razor blade being the most protective CDK components across all delivery locations. Home deliveries benefited from the use of gloves, plastic sheet, and razor blade, while facility deliveries still incurred protective benefits from the gloves, cord clamp, and plastic sheet when adjusted for covariates. Covariates such as newborn male sex and prematurity showed associations with higher rates of newborn mortality. Mothers aged between 20 and 29 years old were less likely to have newborn deaths compared to mothers under 20 years of age (Table 4).

The adjusted logistic regression model for perinatal mortality showed an association between lower odds of perinatal mortality and use of gloves, cord clamps, plastic sheets, and razor blades at all delivery locations. When stratified by delivery location, home deliveries had an association between lower risk of perinatal mortality and use of cord clamps, plastic sheets, and razor blades. Facility deliveries had an association between cord clamp use and reduced perinatal mortality risk. Other covariates' associations with perinatal mortality were similar to the newborn mortality associations. The only exception was birth weight: Low birth weight was not associated with increased perinatal mortality. However, 99.7% of the stillbirths were normal weight (S2 Table).

When danger signs of newborn infection were assessed, 397/36,996 newborns (1%) had infection danger signs at day 1, 141/36,642 (0.4%) at day 4, 153/36,566 (0.4%) at day 10, and 130/36,478 (0.4%) at day 28. Multivariable logistic regression did not show any significant associations between signs of newborn infection at days 1, 4, 10, and 28 and CDK components or any of the covariates.

## Association of timing of death with CDK component use

Unadjusted and adjusted logistic regression models showed a reduced odds of immediate newborn death when gloves (odds ratio [OR] 0.47, 95% CI 0.24–0.91) and cord clamp (0.57, 95% CI 0.32–0.99) were used. Reduction in early newborn death was associated with use of gloves (OR 0.26, 95% CI 0.16–0.39), cord clamp (OR 0.41, 95% CI 0.28–0.60), plastic sheet (OR 0.37, 95% CI 0.24–0.55), and razor blade (OR 0.59, 95% CI 0.40–0.85). In contrast, no individual component of CDK use had a significant association with late newborn death (Table 5). Adjusted logistic regression confirmed similar associations between CDK components and timing of death. The associations between immediate/early newborn death with prematurity, birth weight, and male sex remained ($p < 0.05$).

Cause of death data from verbal autopsy were available for 274 of the 521 newborn deaths. The most common cause overall was newborn sepsis, responsible for 88 (32.1%) of the autopsied newborn mortality, with infectious causes, including pneumonia, tetanus, meningitis, and diarrhea, responsible for 109 newborn deaths (39.8%). In the first 24 hours of life, perinatal asphyxia was the leading cause of death, at 49 (33.8%), with all infections combining for 34 (23.4%) deaths in this period. Between days 1 and 6 of life, infections were the leading cause of death, causing 40 (48.8%) of newborn deaths. Infections caused a further 35 newborn deaths (74.4%) between 7 and 28 days and were the leading cause of death among autopsied newborns.

## Discussion

Our analysis showed that components of CDK were associated with reductions in NMR for use of each component of CDK compared to nonuse. More than 99% of women reported

**Table 4. Adjusted logistic regression—association between CDK components and newborn mortality stratified by delivery location.**

| Variable | All deliveries | | | Home deliveries | | | Facility deliveries | | |
|---|---|---|---|---|---|---|---|---|---|
| | OR | 95% CI | *p*-Value | OR | 95% CI | *p*-Value | OR | 95% CI | *p*-Value |
| CDK component | | | | | | | | | |
| Soap | 0.94 | 0.75–1.16 | 0.56 | 0.69 | 0.46–1.03 | 0.07 | 1.02 | 0.79–1.32 | 0.89 |
| Gloves | 0.33 | 0.24–0.46 | <**0.001** | 0.49 | 0.26–0.93 | **0.03** | 0.38 | 0.25–0.59 | <**0.001** |
| Cord clamp | 0.51 | 0.38–0.68 | <**0.001** | 0.81 | 0.51–1.28 | 0.36 | 0.32 | 0.21–0.51 | <**0.001** |
| Plastic sheet | 0.46 | 0.34–0.63 | <**0.001** | 0.30 | 0.15–0.60 | **0.001** | 0.62 | 0.42–0.90 | **0.01** |
| Razor blade | 0.69 | 0.53–0.89 | **0.004** | 0.31 | 0.17–0.58 | <**0.001** | 0.83 | 0.62–1.13 | 0.23 |
| Candle | 1.32 | 0.57–3.04 | 0.52 | 1.12 | 0.31–4.57 | 0.80 | 1.55 | 0.56–4.31 | 0.40 |
| Matches | 0.62 | 0.27–1.45 | 0.27 | 0.58 | 0.15–2.22 | 0.43 | 0.64 | 0.23–1.82 | 0.40 |
| Maternal age (years) | | | | | | | | | |
| Less than 20 | Ref | | | Ref | | | Ref | | |
| 20–29 | 0.62 | 0.50–0.77 | <**0.001** | 0.57 | 0.38–0.85 | **0.006** | 0.63 | 0.49–0.82 | **0.001** |
| 30–39 | 0.84 | 0.66–1.07 | 0.17 | 0.68 | 0.43–1.07 | 0.09 | 0.92 | 0.69–1.24 | 0.60 |
| 40 or more | 1.16 | 0.76–1.78 | 0.50 | 1.13 | 0.59–2.18 | 0.71 | 1.04 | 0.58–1.91 | 0.88 |
| Maternal education | | | | | | | | | |
| Did not finish primary school | Ref | | | Ref | | | Ref | | |
| Finished primary but not secondary school | 1.11 | 0.91–1.34 | 0.30 | 0.82 | 0.56–1.20 | 0.31 | 1.24 | 0.99–1.55 | 0.06 |
| More than secondary school | 0.52 | 0.13–2.12 | 0.36 | N/A | N/A | N/A | 0.59 | 0.14–2.43 | 0.47 |
| No response | 1.57 | 0.59–4.17 | 0.36 | 1.53 | 0.31–7.42 | 0.60 | 1.88 | 0.52–6.70 | 0.33 |
| Sex of child | | | | | | | | | |
| Female | Ref | | | Ref | | | Ref | | |
| Male | 1.42 | 1.19–1.70 | <**0.001** | 1.44 | 1.05–1.98 | **0.03** | 1.40 | 1.12–1.74 | **0.003** |
| Newborn birth weight | | | | | | | | | |
| Normal | Ref | | | Ref | | | Ref | | |
| Low birth weight | 1.21 | 0.86–1.70 | 0.27 | 1.47 | 0.58–3.76 | 0.42 | 1.24 | 0.86–1.80 | 0.24 |
| Very low birth weight | 2.54 | 0.93–6.96 | 0.07 | N/A | N/A | N/A | 2.74 | 0.98–7.70 | 0.06 |
| Newborn estimated gestational age | | | | | | | | | |
| Term | Ref | | | Ref | | | Ref | | |
| Preterm | 1.75 | 1.41–2.16 | <**0.001** | 2.30 | 1.60–3.30 | <**0.001** | 1.44 | 1.11–1.90 | 0.007 |
| Very preterm | 6.82 | 5.13–9.08 | <**0.001** | 10.62 | 6.84–16.50 | <**0.001** | 5.09 | 3.45–7.52 | <**0.001** |
| Extremely preterm | 9.02 | 6.22–13.07 | <**0.001** | 4.40 | 1.81–10.66 | **0.001** | 11.24 | 7.41–17.07 | <**0.001** |

ORs are adjusted for known risk factors for newborn mortality. Significant *p*-values in bold.

CDK, clean delivery kit; N/A, not applicable; OR, odds ratio.

using at least 1 CDK component. We found a high usage (>80%) of certain CDK components including gloves, cord clamps, plastic sheets, and razor blades. In fact, 81% of deliveries used all 4 of those components. There was a significantly lower odds of newborn mortality individually associated with the use of each of these 4 CDK components, in both home and facility deliveries. Use of gloves and cord clamp were associated with lower odds of newborn mortality in the first 24 hours of life, while use of gloves, cord clamp, plastic sheet, and razor blade were associated with lower odds of newborn death in the first 7 days of life.

There were differences in the CDK components used between home and facility deliveries, with the rates of CDK use at home generally higher than at facilities. This could be due to the presence of alternative equipment at facilities; presence of hand sanitizing supplies and sterilized scissors would exclude the need to use the CDK components soap and razor blade, respectively. Similarly, candles and matches may not be needed if the childbirth occurred during daytime or if the facility had robust electrical infrastructure.

**Table 5. Adjusted association between timing of death and the use of clean delivery kit (CDK) components.**

|  | OR | 95% CI | p-value | NMR Users (deaths/1,000 live births) | NMR Non-users (deaths/1,000 live births) |
|---|---|---|---|---|---|
|  | Immediate death (less than 24 hours) | | | | |
| Soap | 0.85 | 0.58–1.23 | 0.38 | 3.5 | 6.1 |
| Gloves | 0.47 | 0.24–0.91 | **0.03** | 3.8 | 14.9 |
| Cord clamp | 0.57 | 0.32–0.99 | **0.05** | 3.8 | 10.4 |
| Plastic sheet | 0.81 | 0.43–1.54 | 0.52 | 3.8 | 11.9 |
| Razor blade | 0.85 | 0.54–1.35 | 0.49 | 3.7 | 7.7 |
| Candles | 0.35 | 0.05–2. 56 | 0.30 | 2.0 | 5.3 |
| Matches | 1.19 | 0.16–8.82 | 0.86 | 2.1 | 5.2 |
|  | Early death (between 1 and 7 days) | | | | |
| Soap | 0.89 | 0.64–1.22 | 0.46 | 4.7 | 12.3 |
| Gloves | 0.26 | 0.16–0.39 | **<0.001** | 4.8 | 65.1 |
| Cord clamp | 0.41 | 0.28–0.60 | **<0.001** | 4.8 | 37.2 |
| Plastic sheet | 0.37 | 0.24–0.55 | **<0.001** | 4.9 | 49.3 |
| Razor blade | 0.59 | 0.40–0.85 | **0.01** | 4.6 | 21.5 |
| Candles | 1.67 | 0.52–5.29 | 0.39 | 4.8 | 7.7 |
| Matches | 0.59 | 0.18–1.90 | 0.38 | 4.7 | 7.7 |
|  | Late death (between 7 and 28 days) | | | | |
| Soap | 1.12 | 0.71–1.78 | 0.62 | 3.2 | 3.5 |
| Gloves | 0.79 | 0.31–2.07 | 0.64 | 3.2 | 5.7 |
| Cord clamp | 1.02 | 0.47–2.21 | 0.96 | 3.2 | 4.3 |
| Plastic sheet | 0.55 | 0.26–1.18 | 0.12 | 3.2 | 7.0 |
| Razor blade | 0.74 | 0.43–1.30 | 0.30 | 3.1 | 4.8 |
| Candles | 2.46 | 0.62–9.71 | 0.20 | 3.5 | 3.2 |
| Matches | 0.50 | 0.13–2.00 | 0.33 | 3.4 | 3.2 |

*Adjusted with known risk factors for newborn mortality

Despite the guidelines pushing for more facility deliveries in Zambia, lack of equipment at health facilities continues to pose a barrier to clean deliveries [21]. At the time of this study, only half of health facilities in Southern Province had access to soap, and only 3% to sterile gloves [22]. Supplying CDKs may be a low-cost intervention that can overcome this supply challenge for both home and facility births. A review by Bhutta et al. of existing cost-effective interventions to reduce global newborn mortality identified CDKs as an intrapartum intervention for promoting clean delivery practices [23].

Each CDK component associated with lower newborn mortality has a reasonable explanation that aligns with WHO's "six cleans," which doubled as their rationale for their inclusion in the CDK. Gloves improve hand hygiene, sterile cord clamp and razor blade provide clean cord care, and the plastic sheet makes for a clean birthing environment. Seward et al. showed that clean delivery practices associated with lower newborn mortality in Southeast Asia included use of boiled thread to tie the cord, boiled blade to cut the cord, and plastic sheet [12]. The CDK components in our analysis that were associated with reduced newborn mortality corresponded to these practices, providing credence to the mechanism of their benefit.

Our analysis did not find a significant association between CDK use or other covariates and infection danger signs measured in the first 28 days. This discrepancy may have been due to the rarity of reported danger signs, or subjective and inconsistent measurements of the infection danger signs. Possible inconsistencies in danger sign recording include newborn temperature documentation without thermometer confirmation or signs of mild illness mistakenly

documented as signs of life-threatening systemic bacterial infection. It is also possible that if the danger signs were recorded on a postnatal visit after newborn death but still categorized in the same time frame as newborn death—immediate, early, or late—a correlation between the 2 variables would have been compromised. This emphasizes the importance of systemically robust measurements of danger signs in research in global settings.

A major strength of this study is the large sample size, which allowed differentiating between the granular effects of each CDK component. In addition, the trial design focused on the community setting, where a large proportion of women deliver and where newborn mortality from infections remains high but is relatively understudied compared to in major hospitals. This allowed us to study the impact of the interventions in a setting where the participants' choices would mirror their regular behavior, giving us confidence that CDKs may be beneficial whether mothers deliver at home or facilities. The community design also allowed long-term follow-up, with information up to 28 days after the delivery being useful for assessing the impact of CDKs in later infections.

There were a few noteworthy limitations to this study. The use of CDK components was self-reported and collected 4 days after birth, which subjected it to recall bias. Frequent visits by the field monitors were designed to minimize recall bias. The potential that mothers with negative newborn outcomes might not remember their use of CDKs could have led to an inaccurate association between CDK use and newborn mortality. A reverse causality bias is also possible: Childbirth with negative outcomes may require focus on emergent care or aspects other than clean delivery, leading to lower CDK use. Lastly, the original study was not designed to investigate the impact of CDK and thus cannot establish causality for each CDK component. The rate of uptake of CDK was high, and the number of nonusers of each component of CDK was relatively minimal. Nonetheless, the large sample size and differences in NMR support the association between CDK use and reduced newborn mortality in both home and facility deliveries.

While this analysis was based on a dataset from rural Zambia, the mechanism of impact of CDKs may be applicable to other rural settings where home deliveries are prevalent, or health facilities have inconsistent supply of sterile delivery equipment. The effect of CDKs on newborn mortality in the first 7 days should also be generalizable to other settings; although specific microbe composition may vary by location, sterile delivery practices enforced by CDKs would still be effective. For future steps, microbiological examination of newborn infection may be helpful in discerning why there was a greater reduction in infection risk in the first 7 days compared to between 7 and 28 days, and how to reduce newborn mortality in the later newborn stages. In addition, despite evidence that the use of CDKs can also reduce maternal risk of puerperal sepsis, a systematic analysis by Hundley et al. showed that currently there is scant literature on CDKs' impact on maternal morbidity and mortality [24,25]. Finally, a study in Pakistan showed that supplying interventions including CDKs directly to women had varied compliance to the correct use of the equipment and did not reduce newborn mortality [26]. Further studies are required to develop the most effective methods of supplying CDKs to improve adherence to best practices and increase use of components such as cord clamps, which were used less at homes than facilities despite being associated with reductions in risk of newborn and perinatal mortality.

In summary, this analysis demonstrated an association between CDK use and reduction of newborn mortality in a rural setting in sub-Saharan Africa. Utilizing a uniquely large dataset of almost 37,000 newborn outcomes, we found that, among the CDK components, use of gloves, cord clamps, plastic sheets, and razor blades were associated with lower NMR. Our analysis demonstrated a significant association between CDK use and improved newborn

outcomes in the first 7 days of life. Given these positive associations, scaling up CDKs may be an effective strategy to reduce the burden of newborn mortality.

## Supporting information

**S1 CONSORT Checklist.**
(DOC)

**S1 Table. Clean delivery kit (CDK) use by component and perinatal mortality rate.**
(DOCX)

**S2 Table. Adjusted logistic regression—association between clean delivery kit (CDK) components and perinatal mortality stratified by delivery location.**
(DOCX)

**S3 Table. Adjusted association between timing of newborn death and the use of clean delivery kit (CDK) components—with covariates.**
(DOCX)

## Acknowledgments

Research reported in this publication was supported by the Fogarty International Center and National Institute of Mental Health. We would like to thank the ZamCAT field office including the field monitors and the administrative, training, and data management team members, and the members of the data and safety monitoring board and technical advisory group. We are deeply appreciative of the support from the Ministry of Health, Southern Province Medical Office, District Medical Offices, and Chiefs of Southern Province. Finally, we would like to thank the women, their newborns, and their families for their participation and dedication to the study.

The content is solely the responsibility of the authors and does not necessarily represent the official views of the National Institutes of Health.

## Author Contributions

**Conceptualization:** Davidson H. Hamer, Reuben Mbewe, Nancy A. Scott, Julie M. Herlihy, Kojo Yeboah-Antwi, Katherine E. A. Semrau.

**Data curation:** Davidson H. Hamer, Reuben Mbewe, Julie M. Herlihy, Katherine E. A. Semrau.

**Formal analysis:** Jason H. Park, Katherine E. A. Semrau.

**Funding acquisition:** Davidson H. Hamer, Katherine E. A. Semrau.

**Investigation:** Davidson H. Hamer, Julie M. Herlihy, Katherine E. A. Semrau.

**Methodology:** Jason H. Park, Katherine E. A. Semrau.

**Project administration:** Davidson H. Hamer, Reuben Mbewe, Kojo Yeboah-Antwi, Katherine E. A. Semrau.

**Software:** Jason H. Park, Katherine E. A. Semrau.

**Supervision:** Davidson H. Hamer, Katherine E. A. Semrau.

**Visualization:** Jason H. Park, Katherine E. A. Semrau.

**Writing – original draft:** Jason H. Park, Katherine E. A. Semrau.

**Writing – review & editing:** Davidson H. Hamer, Reuben Mbewe, Nancy A. Scott, Julie M. Herlihy, Kojo Yeboah-Antwi, Katherine E. A. Semrau.

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
