## [Editor Report · Decision Letter 0]

15 Dec 2020

Dear Dr Park, 

Thank you for submitting your manuscript entitled "Differential effects of components of clean delivery kits on newborn mortality: results from the Zambia Chlorhexidine Application Trial (ZamCAT)" for consideration by PLOS Medicine.

Your manuscript has now been evaluated by the PLOS Medicine editorial staff and I am writing to let you know that we would like to send your submission out for external peer review.

Kind regards,

Artur A. Arikainen,

Associate Editor

PLOS Medicine

---

## [Decision Letter · Decision Letter 1]

30 Jan 2021

Dear Dr. Park,

Thank you very much for submitting your manuscript "Differential effects of components of clean delivery kits on newborn mortality: results from the Zambia Chlorhexidine Application Trial (ZamCAT)" (PMEDICINE-D-20-05939R1) for consideration at PLOS Medicine. 

Your paper was evaluated by a senior editor and discussed among all the editors here. It was also sent to three independent reviewers, including a statistical reviewer (r#2). The reviews are appended at the bottom of this email and any accompanying reviewer attachments can be seen via the link below:

[LINK]

In light of these reviews, I am afraid that we will not be able to accept the manuscript for publication in the journal in its current form, but we would like to consider a revised version that addresses the reviewers' and editors' comments. Obviously we cannot make any decision about publication until we have seen the revised manuscript and your response, and we plan to seek re-review by one or more of the reviewers. 

We expect to receive your revised manuscript by Feb 22 2021 11:59PM. Please email us (plosmedicine@plos.org) if you have any questions or concerns.

We look forward to receiving your revised manuscript. 

Sincerely,

Emma Veitch, PhD

PLOS Medicine

On behalf of Artur Arikainen, PhD, Associate Editor, 

PLOS Medicine

plosmedicine.org

*Please structure your abstract using the PLOS Medicine headings (Background, Methods and Findings, Conclusions -- "Methods and Findings" is a single subsection). 

*It might be appropriate to add to the abstract (briefly) the context that although conducted within a randomized trial, the randomization was ignored for the purposes of this analysis, with the participants' data pooled across arms (assuming that is correct) - essentially converting it into something akin to a prospective cohort analysis. That could also be spelt out more clearly upfront in the Methods section, potentially. 

*In the last sentence of the Abstract Methods and Findings section, please include a note about any key limitations of the study's methodology - perhaps one of the most obvious relates to the above point, also brought out by reviewers noting that causality cannot be firmly established due to the potential for confounding (which could be expanded on in the main Discussion, as noted by one reviewer, in terms of the potential source(s) and direction of this). 

*Please add the trial registration details to the end of the abstract as per in this (published) example - https://journals.plos.org/plosmedicine/article?id=10.1371/journal.pmed.1002775

*At this stage, we ask that you include a short, non-technical Author Summary of your research to make findings accessible to a wide audience that includes both scientists and non-scientists. The Author Summary should immediately follow the Abstract in your revised manuscript. This text is subject to editorial change and should be distinct from the scientific abstract. Please see our author guidelines for more information: https://journals.plos.org/plosmedicine/s/revising-your-manuscript#loc-author-summary

*Regarding the point above, and raised by reviewer(s), currently the Discussion section notes - "In summary, this analysis demonstrated the impact of CDKs on reducing newborn mortality..." - and it might be appropriate to reword this, particularly use of "impact". 

*Please complete the CONSORT checklist (https://www.equator-network.org/reporting-guidelines/consort/) and include a copy of the completed checklist as supporting material with the revised paper. Ideally please ensure that all relevant parts of CONSORT are described in the revised paper, although some elements will only be relevant (and can be found) in the original trial publication. 

*Please clarify if the analysis reported here corresponds to one laid out in a prospective protocol or analysis plan. Please state this (either way) early in the Methods section.

Comments from the reviewers:

Reviewer #1: 

General comments

Clean delivery kits may reduce death from neonatal infections so this study assessed the potential associations of use of components of the clean delivery kits with newborn mortality in a large cluster RCT. 

This study assesses associations so causality should not be inferred. The title implies causality, which should be corrected. 

Perinatal mortality would be a good measure but stillbirths are not reported. 

Abstract

The abstract should address that the assessment of the use of the components of the CDK by the mothers and was done late. This should caution the interpretation of the results. 

Introduction

The rationale is well-justified. 

The final sentence should not imply causality. 

Methods

The data on use of the components of the clean delivery kits (CDK) was obtained from the mothers four days after delivery. It is unfortunate these data were not collected at delivery because recall bias can severely affect the results. The mothers may not be able to recall details of the use of the CDK. These data were collected after day 1 mortality and most of the first week mortality were known. This can also lead to major biases.

Non-users of specific CDK component was too low to analyze statistically. This is an important limitation which combined with the recall bias and data collection after most of the first week deaths limits the confidence in the results. 

Stillbirth data are needed as stillbirths could be due to infection and because the distinction between stillbirth and death in the first minutes is difficult to assess. It is essential to report the combined stillbirth data with the day 1 mortality data as well as the stillbirth/first week mortality data (perinatal death) which is ideal for reporting when assessing intrapartum and birth interventions.

Results

Infection in an important cause of late neonatal mortality but individual component of CDK use had not significant association with late newborn deaths. Data on use of components of the CDK were collected on day 4 so it is possible that recall bias had nothing to do with late deaths and thus no association. Furthermore, multivariate logistic regression did not show any significant associations between signs of newborn infection at days 1, 4, 10, or 28 and CDK components or any of the covariates. This raises major concerns about the purported association of components of the CDK and neonatal mortality.

It is not reported whether the deaths in the first week disproportionately caused by infections. Data on verbal autopsy and causes of death were collected but not reported. 

Discussion

The Discussion should address the results as potential associations rather than imply causality which is implied multiple times in the Discussion. Terms such as impact, effect, and benefit should not be used in the interpretation of the results. 

Reviewer #2: 

Thanks for the opportunity to review your manuscript. My role is as a statistical reviewer so my comments and queries focus on the study design, data, and analysis. 

This study presents secondary analyses from a large cRCT aiming to reduce neonatal mortality in Zambia. These include comparing neonatal mortality rates according to use of specific components of a clean delivery kit, maternal and neonatal characteristics. There are some very strong differences in rates of NMR with some of the kit components.

I have put general queries first, followed by questions specific to a particular part of the manuscript (with a page and line reference).

Some of the language about the association between the CDK components and NMR strays into causation e.g. "showed reduction in risk" (L217), there are few other instances of this. These should be rephrased as although these relationships are strong and seem reasonable there is still the possibility of confounding with an unmeasured variable. 

I would also try not to use 'significant' or 'statistically significant', rather focus on the effect estimates and use the p-values to as a demonstration of evidence. The effect sizes are very impressive so it would be a shame not to highlight these.

P6. L164. To confirm, the main study used GEEs as there was cluster randomisation, the randomisation status wasn't used in any of these analyses so a standard logistic regression model was used throughout? 

P6, L167. 'Confounding by indication' has become a fairly vague term with different interpretations in different sub-fields of health and medicine. It would be more useful to be specific about how you think the confounding would affect your analysis here (keeping in mind PLOS Med has a general audience). 

P6, L174. Theoretically there are 2**7 = 128 permutations of use of the 7 different components of the CDK. How were the 'key permutations' decided? It isn't clear what the 'analysis of interactions' was - is this the interaction with the other patient and cluster level characteristics?

P6, L175. Were the risk factors for newborn sepsis that were included as adjustments used in the same form as presented in table 1?

P6, L180. Was an analysis plan developed for this secondary analysis? 

P6, L189. The missing data strategy (complete-case analysis) should be detailed in the methods. It would also be helpful to see basic summary information that compares participants included vs. excluded on key characteristics (e.g. what is presented in Table 1). I don't think that a complete-case excluding ~2.5% of the original participants is likely to be a problem, but it would be reassuring to see there were no strong differences.

P9, L238. Was the interaction between delivery location and CDK etc. formally tested with an interaction?

Reviewer #3: 

While this study is a secondary analysis that employs data from the primary Zambia chlorhexidine Application Trial, the findings are nonetheless compelling. 

Almost 37,000 newborns were included in the analyses which reported on independent rates of the "6 cleans" based upon the provision of clean delivery kits. Independent rates of reduction in mortality for each component contained in the kit have been reported, along with its association on immediate newborn death, early as well as late neonatal mortality.

It appears evident from data provided in this paper, utilizing logistic regression analyses, that employment of clean birthing practices has the potential to produce significant reductions in mortality. The impact appears to be meaningful immediately after birth and within 7 days of delivery with no additional reduction in mortality after that time. 

The authors appropriately point out the weaknesses inherent in using self-reported data. It would be interesting to determine why cord clamping was not practiced and home births relative to its use at facilities. Suggestions for education and training of community health providers may be important in addressing this difference in future interventions.

The paper would be enhanced by including costs associated with providing the intervention. 

Overall, however, there is sufficiently strong evidence presented to expand the use of clean delivery kits as an important component for reducing immediate and early neonatal mortality in Zambia and likely in other sub-Saharan communities.

[LINK]

---

## [Decision Letter · Decision Letter 2]

24 Mar 2021

Dear Dr. Park,

Thank you very much for re-submitting your manuscript "Association between components of clean delivery kits and newborn mortality: results from the Zambia Chlorhexidine Application Trial (ZamCAT)" (PMEDICINE-D-20-05939R2) for consideration at PLOS Medicine.

I have discussed the paper with editorial colleagues and our academic editor, and it was also seen again by one reviewer. I am pleased to tell you that, provided the remaining editorial and production issues are dealt with, we expect to be able to accept the paper for publication in the journal.

[LINK]

Please let me know if you have any questions, and we look forward to receiving the revised manuscript shortly.   

Sincerely,

Richard Turner, PhD

rturner@plos.org

Requests from Editors:

Please finalize the arrangements for data deposition in a publicly-accessible repository. 

So as to comply with journal style, please adapt the title to "Components of clean delivery kits and newborn mortality in the Zambia Chlorhexidine Application Trial (ZamCAT): an observational study" or similar. 

Please quote the study dates in the abstract.

At lines 44-49, please adapt "reduced newborn mortality" to "lower risk of newborn mortality", for example.

Please add a new final sentence to the "Methods and findings" subsection of your abstract, which should begin "Study limitations include ..." or similar and quote 2-3 of the study's main limitations. 

Please use the active voice for one or two points in your Author summary (e.g., "We analysed data ..."). 

Please state explicitly in your Methods section that the analysis did not have a protocol or prespecified analysis plan (assuming this is the case); are you able to state that there were no data-driven changes in the analysis plan?

Please avoid language implying causality throughout the text. For example, at line 400 rather than "were associated with the greatest reduction ..." please adapt the text to "were associated with the lowest risk of ..." or similar. 

Please use the following style throughout the text, except where a number is used to start a sentence: "All 7 components ...". 

Please remove the information on study funding from the Acknowledgements section at the end of the main text. This information should appear only in the article metadata, via entries in the submission form. 

Please adapt the attached CONSORT checklist so that individual items are referred to by section (e.g., "Methods") and paragraph number rather than by page or line numbers, as the latter generally change in the event of publication. 

Please rename the file "S1_CONSORT_Checklist" or similar and refer to it in your Methods section by this label.

Comments from Reviewers:

*** Reviewer #2: 

Thanks for the revised manuscript and replies to my original queries. This is an interesting and useful manuscript and overall I recommend it. There are a few minor changes I recommend.

The language around causality is improved. In L401 I think 'connection' is still too strong, 'association' would be more accurate. 

For the most common combinations, I would add change the table description from 'top' to 'most frequent' as this is more specific and not every reader will be examining the methods as closely as I like to.

 Women excluded from the analysis for missing data are broadly the same as those included - there are more older women, and more likely to deliver in a hospital, but I don't see this as being problematic with 1583 missing from the original sample. 

L297. 'Multivariable' is a more suitable description with one outcome and several covariates.

L308. Is the adjusted analysis shown somewhere? If in an appendix (I couldn't find it) please refer to this. This should be included somewhere, either by expanding Tab 4 or including in an appendix.

Supplemental Table 1 didn't appear in the document when I downloaded it.

***

[LINK]

---

## [Editor Report · Decision Letter 3]

3 Apr 2021

Dear Dr Park, 

On behalf of my colleagues and the Academic Editor, Dr Myers, I am pleased to inform you that we have agreed to publish your manuscript "Components of clean delivery kits and newborn mortality in the Zambia Chlorhexidine Application Trial (ZamCAT): an observational study" (PMEDICINE-D-20-05939R3) in PLOS Medicine.

Prior to final acceptance, please remove the grant number from the acknowledgements section (this also appears in the article metadata). 

Please also add details of the page providing access to study data (in article metadata). 

PRESS

Sincerely, 

Richard Turner, PhD 

rturner@plos.org